# Physico-Mechanical, Thermal, Morphological, and Aging Characteristics of Green Hybrid Composites Prepared from Wool-Sisal and Wool-Palf with Natural Rubber

**DOI:** 10.3390/polym14224882

**Published:** 2022-11-12

**Authors:** Seiko Jose, Puthenpurackal Shajimon Shanumon, Annmi Paul, Jessen Mathew, Sabu Thomas

**Affiliations:** 1School of Chemical Sciences, Mahatma Gandhi University, Kottayam 686560, Kerala, India; 2Textile Manufacturing and Textile Chemistry Division, ICAR-Central Sheep and Wool Research Institute, Avikanagar 304501, Rajasthan, India; 3School of Energy Materials, Mahatma Gandhi University, Kottayam 686560, Kerala, India

**Keywords:** aging, coarse wool fibre, hybrid composites, natural rubber, PALF, sisal fibre

## Abstract

In the reported study, two composites, namely sisal-wool hybrid composite (SWHC) and pineapple leaf fibre(PALF)-wool hybrid composite (PWHC) were prepared by mixing natural rubber with equal quantities of wool with sisal/PALF in a two-roll mixing mill. The mixture was subjected to curing at 150 °C inside a 2 mm thick mold, according to the curing time provided by the MDR. The physico-mechanical properties of the composite *viz*., the tensile strength, elongation, modulus, areal density, relative density, and hardness were determined and compared in addition to the solvent diffusion and thermal degradation properties. The hybrid composite samples were subjected to accelerated aging, owing to temperature, UV radiation, and soil burial tests. The cross-sectional images of the composites were compared with a scanning electron microscopic analysis at different magnifications. A Fourier transform infrared spectroscopic analysis was conducted on the hybrid composite to determine the possible chemical interaction of the fibres with the natural rubber matrix.

## 1. Introduction

Green composites are expected to be the next generation of sustainable composite materials, and both academia and industry are interested in them [1]. These materials are made from natural resources that are renewable, recyclable, and biodegradable. Green composites are typically made by combining natural resins with plant and animal fibres. Natural fibres are demonstrating that they are a more ecologically friendly, cost-effective, and a lighter alternative to synthetic fibres [2]. Bio-resins, which are derived from protein, starch, and vegetable oils, have been created as an alternative to petroleum-based polymers. Compared to synthetic fibres, natural plant-based fibres have a number of clear advantages, such as a reasonable price, good mechanical properties, thermal and acoustic insulation, and can degrade naturally. The short natural fibre reinforced rubber composites have been found to possess a good dimensional stability and high green strength [3].

Sisal is a commercially valuable fibre, extracted from *Agave sisalana* leaves. It is primarily utilized in the production of carpets, insulating panels, and maritime ropes and is commercially grown in Brazil, Tanzania, Kenya, and Madagascar. It has tremendous tensile strength and is quite robust. Many research attempts have been reported for the use of sisal fibre in composites [4]. Pineapple is one of the most popular fruits extensively grown in Costa Rica, the Philippines, Brazil, Thailand, China, and India. The pineapple leaf fibre (PALF) is extracted from the leftover leaves of the pineapple plant. Of all of the natural fibres derived from plant leaves, PALF has the largest proportion of cellulose content and the lowest microfibrillar angle, which results in an exceptionally good tensile strength. PALF isused for a variety of purposes, such as the creation of textiles [5], paper [6], and composites [7]. Both PALF and sisal fibres are extracted by a mechanical extraction process known as decortication.

Wool is a protein fibre obtained from sheep. Based on the fibre fineness, it is categorized as fine, medium coarse and very coarse (kempy). The fine and medium coarse wool is employed in the apparel and carpet industries, respectively. The very coarse wool fibre has medullation, as result it is highly brittle and does notfind applications in the above said industries. Currently, they are used in decentralized sectors, *viz*., quilt industries, handmade felt, and so forth [8]. Few studies have been reported about the use of coarse wool in composites [9,10,11].Natural rubber (NR) is one of the unavoidable polymers in the world. It is extracted as a resin from the *Hevea brasiliensis* tree and further dried into sheets or blocks. The plastic qualities of the natural rubber are transformed into elastic through vulcanization, ultimately resulting in the hardness and resilience of NR. The NR is a highly preferred matrix for the composite researchers.

Several recent studies have been reported, regarding the fabrication of PALF and sisal fibre reinforced natural rubber composite [12,13]. Sisal fibre is considered as an important reinforcement, due to the presence of excess cellulose components which make them less susceptible to moisture [14]. The physical and mechanical characteristics of the hybrid composites are determined by factors, such as type of fibre, aspect ratio, orientation, length, and interfacial adherence to the matrix [15].

The objective of our work is to give a value addition to the highly coarse wool, which has no other purpose at the moment. In our previous work, we employed coarse wool fibre as reinforcement in the rubber matrix and subsequently made few prototypes. However, we realized the need of improving the mechanical properties of the wool- NR composite, without compromising the “natural touch”. Thus, it is decided to mix coarse wool with appropriate plant fibre and to prepare the hybrid composites with better mechanical properties. As per the author’s knowledge, the hybrid composite of wool with other plant fibres in the natural rubber matrix, has never been reported. Hybrid composites are often prepared to conceal the flaws of one or more component fibres. Many research attempts have been reported on the hybrid composites of natural fibres [16,17]. In this study, two hybrid composites (wool + sisal + rubber) and (wool + PALF + rubber) were prepared by mixing equal quantities of wool with sisal/PALF in a rubber matrix. The morphological, thermal, physico-mechanical properties, and accelerated aging of these hybrid composites were analyzed and compared.

## 2. Materials and Methods

The natural rubber of grade ISNR-5 (Indian Standard Natural Rubber) was sourced from M/S Malankara Plantations, Thodupuzha, Kerala, India. The coarse wool fibre (Patawadi sheep breed) (bundle strength—13.67 g/tex, fibre diameter—44 microns), sisal fibre (bundle strength—30.9 g/tex, fineness—30 tex, density—1.45 g/cm^3^) was supplied by ICAR- Central Sheep and Wool Research Institute, Avikanagar, Rajasthan, India, and ICAR- Sisal Research Station, Odisha, India, respectively. The PALF was extracted from remnant pineapple leaves after cultivation, using a decorticator at Mahatma Gandhi University, Kottayam, Kerala, India. (bundle strength—38.5 g/tex, fineness—3.5 tex, density—1.43 g/cm^3^). The chemicals used in the rubber vulcanization *viz*., sulphur (sp. gravity 2.05), zinc oxide (sp. gravity 5.55), stearic acid (sp. gravity 0.85), Wingstay L (antioxidant), and CBS (N-cyclohexylbenzothiazylsulphenamide), were purchased from Sameera Chemicals, Kottayam, Kerala, India.

### 2.1. Preparation of the SWHC and PWHC 

The wool, sisal, and PALF fibre were chopped in to 1.5 cm length. The NR was adequately masticated in a two-roll mixing mill (300 × 500 cm) for two minutes. The masticated rubber, the vulcanizing agents, and the fibreswere combined, as mentioned in Table 1. Just before adding sulphur, the wool fibre was added to the NR polymer matrix, along with the sisal fibre for the SWHC and withthe PALF for the PWHC. Care was taken to preserve the compound flow direction so that the majority of the fibres followed the same flow path. In order to ensure an equal distribution of thefibres in the polymer matrix, samples were milled for 10 min [18].

Using a Moving Die Rheometer (Rheometer MDR 2000, Alpha technology), the curing properties of the SWHC and the PWHCwere studied, in accordance with the ASTM D5289 method at 150 °C. The compositeswerevulcanized at 150 °C inside a 2 mm thick mold, according to the curing time provided by the MDR at 100 bar pressure (5 min for curing for both composites (see t_90_ vales in Table 2). Following thevulcanization, the hybrid composite samples were removed from the mold and cooled. The samples were pre-conditioned at 25 °C and 65% RH before further analysis. For each set of composites, fivereplicas were prepared. The cure rate index, which is a measurement of difference between t_90_ (optimum cure time)and ts_2_ (insipient scorch time), was calculated using the formula [21]
(1)$$Cure\;rate\;index=100/\left(t90-ts2\right)$$

### 2.2. Analysis of the Physico-Mechanical Properties of the Composites

A universal testing machine (Tinius Olsen H50KT) was employed for the determination of the tensile and tear strength of the SWHC and PWHC. The samples were analyzed, in accordance with the ASTM D412 and ASTM D624 standards, respectively. Three samples were tested for each composite and the average result was calculated. The moisture content of the hybrid composites was determined, in accordance with ASTM D2495-07. The hardness of the SWHC and PWHC was assessed with the aid of a Shore-A hardness tester (Presto), following the ASTM D-2240 guidelines. The areal density of the composites was calculated, using the following formula.
(2)$$Areal\;density=\frac{Weight\;of\;the\;composite\;\left(g\right)}{Area\;of\;the\;composite\;\left(cm^{2}\right)}\times 10000$$

The relative density of the SWHC and PWHC in water was calculated according to ASTM D792, using the equation below.
(3)$$Relative\;density=\left(\frac{Weight\;in\;air}{Weight\;in\;air-Weight\;in\;water}\right)\times Density\;of\;water$$

The SWHC and PWHC were analyzed for their solvent diffusion properties using water and toluene. Three replicas from each composite were cut in a round disc shape (2 cm diameter).Prior to dipping in the solvent, both specimens underwent preconditioning (25 °C and 65% RH) and were weighed. The hybrid composite samples were dipped in their respective solvents and removed from the solvents in predefined intervals. The specimens were gently hand pressed in between a blotting paper, to remove the surplus solvent and weighed. The procedure was repeated until a swelling equilibrium was reached [18].The mole% solvent uptake of the composite samples was calculated using the Equation (4).Qt represents the solvent’s mole% uptake at a certain time t. Further, to investigate the diffusion properties of the SWHC and PWHC with water and toluene, a graph of Qt vs √t was generated.
(4)$$\mathrm{Qt}=\left(\frac{\left(Mass\;of\;solvent\;absorbed\;by\;thecomposite/Molar\;mass\;ofthe\;solvent\right)}{Initial\;mass\;of\;the\;composite}\right)\times 100$$

The crosslink density, associated with the composites immersed in toluene is calculated using the following sets of equations below [18]
(5)𝞄=1/2Mc
(6)$$Mc=\frac{\rho_{p}\;\mathrm{Vs}\;\phi ⅓}{\left(\ln\;\left[1-\phi \right]+\phi +{𝛞\phi }^{2}\right)}$$
(7)$$\phi =\frac{\frac{W1}{\rho_{p}}}{\left(\frac{W1}{\rho_{p}}\right)+\left(\frac{W2}{\rho_{s}}\right)}$$
(8)$$\chi =\;\beta +\frac{\mathrm{Vs}}{\mathrm{RT}}\;{\left(\delta p-\delta s\right)}^{2}$$

The crosslinking density of the material is given by Equation (5). Equation (6)is used to compute the molar masses between the crosslinks, or “*Mc*”. Equation (7)is used to compute “ϕ”, which is the volume fraction of rubber at the swelling equilibrium. “ρ_p_” stands for the polymer density, “ρ_s_” for the solvent density, and “Vs” for the molar volume of each solvent. Equation (8)can be used to calculate “χ” which is the interaction parameter between the polymer and the solvent. Equation (6) would be used to derive “Mc”, using the values of “ϕ” and “χ”, as determined by Equations (7) and (8), respectively. In Equation (4), “β”“*δs*”, and “*δp*” stand in for the lattice constant (zero for polymers), the solubility parameter of the solvent, and the solubility parameter of the polymer, respectively. “R” is the universal gas constant and “T” is the temperature. For all testing, five replicas were made and the average value was taken.

### 2.3. FTIR, SEM, TGA, and the Aging Analysis of the Composites

With a Perkin Elmer Spectrum-2 spectrometer, the FT-IR spectra of theSWHC and PWHC were recorded across a range of 4000 cm^−1^ to 400 cm^−1^, using the attenuated total reflection (ATR). The spectrum was obtained after 24 consecutive scans. The JEOL-JSM-6390 scanning electron microscope was used to examine the surface morphological properties of the hybrid composites. The samples were sputter coated with gold-palladium to prevent electron beams from having any charge effects during the examination. The images were captured at various magnifications at a 20 kV accelerating voltage. The thermogravimetric analysis was carried out, using TA instruments (SDT Q600) in an inert atmosphere at temperatures ranging from 25 °C to 700 °C. The heating rate, 10 °C/min and a DTA sensitivity of 0.001 °C were maintained throughout the analysis. The composite samples were subjected to accelerated aging to temperature (ASTMD 573-04), UV, and biodegradation as per the standard methods reported in our previous studies [22].

## 3. Results and Discussion

### 3.1. Cure Characteristics

Figure 1 shows the cure characteristics of the SWHC and PWHC. It is apparent from the figure that both composite mixtures followed almost the same pattern in the MDR curve. It can be clearly understood from the graph that the time required for the initiation of the crosslinking is nearly the same for the SWHC and PWHC. There is an initial decrease in torque that was noted in both the SWHC and PWHC, because of the softening of the rubber polymer matrix, when subjected to heat. When the crosslinking was initiated, with respect to time, the torque increased to a maximum, where the crosslinking was the highest and then showed a slight reduction, and then became almost constant. The corresponding torque and cure time values are shown in Table 2.

In fibre reinforced rubber composites, the maximum torque is an indication of the extent of the crosslinking and the stiffness, while the minimum torque indicates the fibre content present in it [23]. The maximum torque for the PWHC (48.68) is marginally higher than that for the SWHC (38.36). The value of ts_2_ (insipient scorch time) is same for both composites, which indicate that the vulcanization in both composites begins at the same time and the similar value of t_90_ shows that the vulcanization proceeds to a completion at the similar time, for the composites.

Regardless of the same values for t_90_ and ts_2_ in both the SWHC and PWHC, the maximum torque for the PWHC was found to be high. This indicates that the crosslinking associated with the vulcanization was increased by the addition of PALF in the wool-NR matrix, in comparison to the addition of the sisal fibre. The cure rate index is almost same for both composites, which indicates that the rate of curing is almost the same for the SWHC and PWHC [24].Since the rubber content is lower in the NR hybrid composite, the cure curve declined after reaching t_90_. This might be due to the over curing of the composites. Similar observations have been reported elsewhere [25].

### 3.2. Tensile and Tear Properties

The stress–strain curve of the composite samples is depicted in Figure 2. The curves indicate that the PWHCs could withstand more stress in comparison with SWHC, meanwhile the latter possess a higher elongation. The corresponding tensile strength data is shown in the Table 3.

It is evident from Table 3 that the tensile strength for the PWHC (11.14 MPa) is almost double thanthat for the SWHC (6.09 MPa). The higher tensile strength of the PWHC, in comparison with the SWHC may be due to the following reasons. (i) The higher tensile strength of the PALF (38.5 g/tex) than sisal fibre (30.9 g/tex), (ii) the dense packing of the fibre, the lack of voids and a better interfacial adhesion of the PALF in the rubber matrix, as observed from the SEM images, (iii) the better transfer of stress between the NR and PALF, compared to that between the NR and sisal fibre, as indicated by the tear analysis (Table 3). Thus, Young’s modulus associated with the PWHC is higher than that of the SWHC. The lower value of Young’s modulus and the elongation (%) indicates that the PWHC has a much better resistance to elastic deformation, compared to the SWHC.

The tear strength of a polymer indicates the ability of the polymer to withstand tearing or cracking, when they are subjected to an external force. The tear strength of the PWHC (85.0 N/mm) is significantly higher than that for the SWHC (45.9 N/mm). This is because the transfer of stress in the PALF incorporated composite, is better than that in the sisal fibre incorporated composite. The lower tear strength is also an indication of a poor interaction between the fibre and the polymer matrix [26]. The SWHC possessed a higher elongation at break (5.42%) than the PWHC (4.49%).

### 3.3. Moisture Absorption and the Hardness Properties

In comparison with the synthetic fibre composites, the moisture content of the SWHC and the PWHC was found to be in a higher range (Table 4). The higher moisture content of the developed hybrid composites may be due to the high inclusion (100 phr) of hydrophilic natural fibres. In comparison with the SWHC, the PWHC has a marginally lower moisture content (5.90%), perhaps due to the fact that sisal fibres absorb more moisture than the PALF [27]. In the context of composites, the high moisture content of the natural fibre is a major concern to researchers. Sisal and PALF are lignocellulosic, and the presence of hemicelluloses causes a high moisture uptake. The presence of moisture in the natural fibre reinforced composites causes a weaker interaction between the fibres and the matrix [28].

Rubber is a soft polymer. The inclusion of natural fibres, such as wool, sisal, and PALF, in large quantity significantly increase the hardness of the composites. A good network formation of a natural fibre inside of the soft rubber polymer matrix may be the reason behind it, as a result, the Shore A hardness increased. The hardness of the PWHC was 91.56 and that of the SWHC was 91.3. Though there exists a considerable difference in the mechanical properties, hardness (91), areal density (2700 g/m²), and relative densities (1.11 g/cm^3^) of both, the PWHC and SWHC were found to be almost the same. It can be seen from Table 4 that the relative density of both hybrid composites is almost same.

### 3.4. FTIR Analysis

Figure 3a displays the FTIR spectra of sisal, PALF, and wool fibre. The sharp peak for wool fibre, seen at 1640 cm^−1^, is due to the amide I group present in the wool protein, while the peak at 3272 cm^−1^ denotes the amide N-H stretching vibration. The bending vibration of the C-N-H bond corresponds to the peaks at 1520 cm^−1^ [29,30,31]. The FTIR spectra shows similar peaks for both the PALF and sisal, as theyare lignocellulosic fibres. In case of the sisal fibre and PALF, the broad peak at 3332 cm^−1^ corresponds to the -OH stretching vibrations from cellulose while the symmetric and asymmetric stretching of the CH_2_ groups is indicated by the peak at 2886 cm^−1^ [32]. The peaks at 1732 cm^−1^ can be attributed to the stretching vibration of C=O groups in hemicellulose and the peaks in between 1627–1606 cm^−1^,indicate the aromatic C=C stretching vibrations in lignin [33,34]. It can also be observed that the intensity of the peak at 1606 cm^−1^ is higher for the sisal fibre, compared to PALF, indicating that the lignin content is higher for the sisal. The C-O/ C-C group stretching is indicated by the peaks at 1017 cm^−1^ [35].

It can be clearly observed, from the Figure 3b, that the SWHC, PWHC, and the vulcanized rubber shows almost the same peaks. The peaks observed at 2920 cm^−1^ and 2848 cm^−1^ indicate the symmetrical stretching of the -CH_3_ bonds and -CH_2_- bonds, respectively. The characteristic in-plane bending of the amide II group in the wool protein is denoted by the peak at 1536 cm^−1^. The sharp peaks observed at 1452 cm^−1^ and 1370 cm^−1^ indicate the deformation of the -CH_3_ bonds, while the out of plane bending for the C=C-H group is shown by the peak at 830 cm^−1^ [36]. All of these peaks are characteristic of the vulcanized rubber sample. It is observed from the spectra that the peaks corresponding to the wool, PALF, and sisal are not prominent and are masked by the natural rubber. No shifts in the peaks, even after the addition of the fibres, indicates that there is no chemical interaction between the fibres and the polymer matrix, leading to the conclusion that the interaction may be physical, involving van der Waals forces or hydrogen bonds.

### 3.5. SEM Analysis

Figure 4a–d shows the cross-sectional scanning electron microscopic images of the SWHC and Figure 4e–h shows that of the PWHC. It is visible from the images that in both the composites, a good network of fibres is present throughout the polymer matrix. It can also be seen from the images that the wool fibre (with scales) is distributed evenly with the sisal fibre (without scales) in the SWHC and with PALF (without scales) in the PWHC. Most of the fibres are in a uniform direction inside the rubber matrix. The absence of large voids in the composites indicates that the hybrid composites are properly prepared without the entrapment of air inside them. However; while comparing the SEM images (Figure 4d,h), which is of the cross sectional images of the SWHC and PWHC, respectively, it is found that the number of voids are comparatively higher in the SWHC than in the PWHC. This indicates a good adhesion between the PALF and NR matrix. The fibre pull-out is visible from the matrix in Figure 4b,f, and the images suggest that in the PWHC (Figure 4h), the voids formed at the root of the pulled-out fibres, are relatively small when compared to that in the SWHC. The higher interfacial adhesion between the natural fibres and the polymer matrix keeps the fibres intact with the matrix, which can also be the reason for the higher tensile strength, shown by the PWHC (11.14 MPa), compared to the SWHC (6.09 MPa) [37]. It has been reported that the formation of large voids is an indication of the poor interfacial adhesion between the fibres and the matrix [13].It is inferred from the SEM analysis that the failure mechanism in these composites was the fibre pull-out, the fibre fracture, and the interfacial debonding.

### 3.6. Thermogravimetric Analysis

Figure 5a shows the TG curve of the SWHC and PWHC. Due to the similarity in the chemical nature and composition, both composites followed almost the same pattern. As discussed regarding the physical properties of the composites, the composites have a moisture content of 6–7%. The minor weight loss at 110 °C, may be due to the removal of moisture from the composites. The TGA shows a major weight reduction between 250 °C and 400 °C and both composites showed a weight loss of 84.14%, until a constant weight was reached at a temperature of 442.5 °C.

The thermal degradation of the composites may be explained, based on the degradation of the individual components. In the case of the wool fibre, the thermal degradation takes place in three steps [38]. The first stage of the degradation takes place between 100 °C and 135 °C and is attributed to the loss of moisture content [39]. During the second step, a maximum weight loss occurred between 218 °C and 390 °C, due to the breakdown of the microfibril-matrix structure and the disulfide linkages [38]. In the third step, various peptide bonds present in the wool were broken at around 390–50 °C. Above 500 °C, the char oxidation reactions dominated [31,40].

Being lignocellulosic in nature, in both the sisal and PALF, after the removal of moisture at 110 °C, the second weight loss corresponds to hemicellulose’s degradation that starts at about 190 °C. Further, cellulose starts degrading from 290 °C up to 360 °C. The lignin degradation starts at about 280 °C and continues even above 500 °C [32]. For the vulcanized rubber, the degradation begins at about 200 °C and is completed at about 475 °C, where the maximum weight loss is obtained at 358 °C, which may be attributed to the oxidation of the rubber [41]. It is also inferred from the data that the incorporation of wool, sisal, and PALF, slightly increases the thermal stability of the vulcanized rubber.

The degradation process has demonstrated one corresponding weight-loss peak in the DTG curves, as shown in Figure 5b, which corresponds to a single turn in the TG curves and was caused by thermal scissions of the C-C chain bonds in the natural rubber matrix [42].The DTG curves show that at 361.67 °C, both composites show an equal rate of weight loss (0.8373%/°C), with respect to the temperature. The results indicate that the SWHC and PWHC possess a similar range of magnitude when considering their thermal stability, which may be due to the fact that both sisal fibre and PALF are plant fibres that havean almost similar chemical structure and properties.

### 3.7. Solvent Diffusion

The uptake of toluene and water by the SWHC and PWHC, via the diffusion, was analyzed and plotted between Qt (mole% uptake of solvent) and √t (min). The process of diffusion is a parameter for the kinetics and is related to the nature of the polymer, the nature of the fillers added, its free volume, the extent of crosslinking, etc. [18]. It is apparent from Figure 6a that the rate of diffusion and the quantity of the toluene absorption is higher in the SWHC, in comparison with the PWHC. The low absorption and diffusion of toluene in the PHWC may be due to the dense packing of PALF inside the rubber matrix, which restrict the diffusion of the aromatic solvent [43]. This is also supported by the SEM images, which showed a higher packing density, a better adhesion, and less void contents in the PWHC than the SWHC. At the time of saturation, the SWHC showed a weight gain of 184.65%, while it was 95.85% for the PWHC.

Interestingly, it can be also seen from Figure 6b, that the mole% uptake of water does not show much difference for both the SWHC and PWHC, although the SWHC graph shows a marginally higher uptake of water. Both composites showed similar rates of diffusion up to the saturation. The water may enter the composite though the small cracks and pores and generates diffusion pathways. Both the sisal fibre and PALF are hygroscopic and allow the diffusion of water through them whilst, the matrix is hydrophobic. At the time of saturation, the SWHC gained 14.99% and the PWHC gained 14.39% weight, respectively. These high water absorption properties, though common in natural fibre-reinforced composites, are not in an appreciable quality for the composites.

It can also be observed from Table 5 that both composites possess a similar crosslink density. The crosslink density was defined as the density of chains or segments that connect two infinite sections of the polymer network [44]. The value of “Mc”, which is the molar masses between the crosslinks, is so high that it can be considered as an indication of the greater crosslinking of the networks present in the composites [44].

### 3.8. Accelerated Thermal and UV Aging

The composites of the NR are susceptible to degradation by heat, UV radiation, ozone, humidity, etc. [45]. The PWHC and SWHC (Figure 7) were subjected to the accelerated thermal and UV degradation and the change in their mechanical properties were analyzed. In large chain macromolecules with complicated crosslinked structures, the application of heat, as well as radiation, can cause scissions, not only to the main chain, but also to the side chains which may lead to a loss of weight and the emission of gases with low molecular weights. As a result, the exposure to heat/radiation can cause changes to the chemical structure of the composites, such as the chain scission, crosslink formation, and breakage [46].It can be observed from Figure 8 that there is a slight increase in the tensile strength and Young’s modulus after the thermal aging for both the SWHC and PWHC. This increase might be due to the formation of new crosslinks when the vulcanized NR is subjected to heating [47]. When exposed to prolonged UV radiation, the tensile strength increased for both the SWHC and PWHC, although there was a significant reduction in Young’s modulus of the material. The trend shown by the un-aged, thermal aged, and UV aged samples for both the PWHC and SWHC, was similar, such that there is an increase in their tensile strengths. In the case of their stress % (elongation at break %), the two composites showed a different trend, such as in the SWHC, the stress % demonstrates an increase after the UV aging, with respect to the un-aged samples, where the stress % for the PWHC remains stagnant. Similarly, Young’s modulus for both the PWHC and SWHC showed a similar trend, such as an increase in the modulus after thermal aging and a decrease in the modulus after UV aging.

### 3.9. Biodegradation

Biodegradation is a process in which a compound decomposes due to the enzymes or chemicals secreted by bacteria or fungi present in soil. Both the SWHC and PWHC were subjected to accelerated biodegradation for 60 days through a soil burial test, as mentioned earlier. Once the stated period was over, the reduction in weight for the SWHC was found to be 2.43%, while it was 2.34% for the PWHC. It can be seen from Table 6 that both composites showed an almost equal loss of weight. It appears that the biological decomposition occurred at a very slow rate. This may be due to the following reason. (1) The vulcanized rubber, as it is poorly biodegradable, due to the presence of high crosslinking. (2) Though wool, sisal, and PALF are natural fibres that degrade over time, the presence of lignin in the PALF and the sisal mask and protects the cellulose and hemicellulose from a rapid degradation by the microorganisms, due to the presence of the aromatic and crosslinked structure of the lignin [13]. (3) The extensive packing of fibres in the matrix can slow down the degradation process, since the tight network prevents the excessive growth of microorganisms [48].Above all, these facts and the presence of high amounts of natural fibres in the developed composites resulted in an increase in the absorption of moisture which eventually led to the growth of microorganisms that caused the mass degradation of the composite.

## 4. Conclusions

Hybrid green composites, SWHC (sisal fibre + coarse wool fibre + NR) and PWHC (PALF + coarse wool fibre + NR) were fabricated and compared for their morphological, physical, mechanical, and aging properties. In comparison with the SWHC, the PWHC showed a higher tensile strength and modulus, a higher tear strength, a low moisture absorption and low mole% uptake (diffusion) of toluene and water. The PHWC showed a higher torque during the cure analysis. The results obtained from the FTIR spectraprovided no valid evidence for any chemical interaction between the polymer matrix and the natural fibres in both the SWHC and PWHC. An analysis of the SEM images showed that the PWHC had a better packing of fibres, thereby an increased interfacial adhesion between the fibres and the polymer matrix. The thermal degradation characteristics remained as constant. Both the hybrid composites showed a slow pace of degradation while subjected to the soil burial test. It is concluded that the newly developed hybrid composites can be regarded as good substitutes for non-biodegradable composites and can be considered as potential material for packing and household applications.

## Figures and Tables

**Figure 1 polymers-14-04882-f001:**
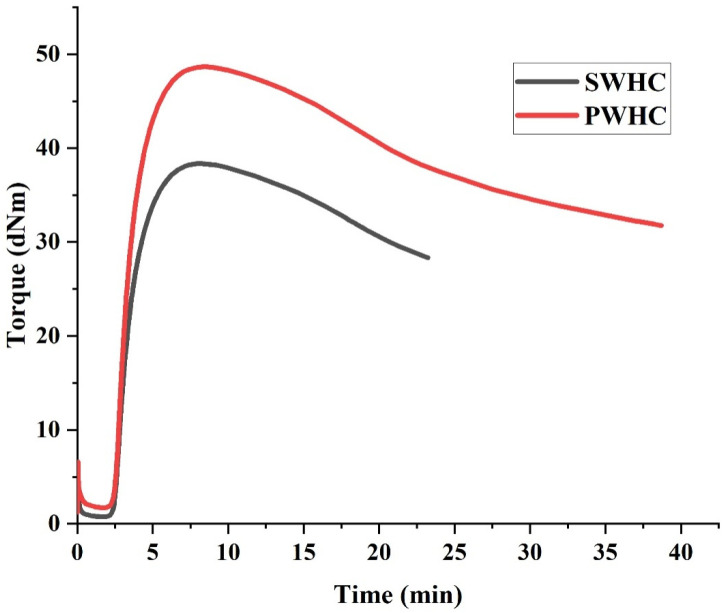
Cure characteristics of the SWHC and PWHC.

**Figure 2 polymers-14-04882-f002:**
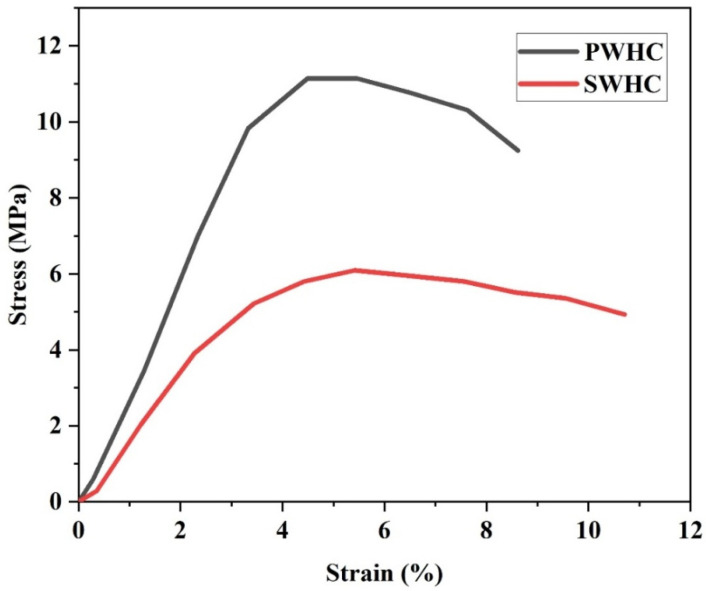
Stress–strain curve of the SWHC and PWHC.

**Figure 3 polymers-14-04882-f003:**
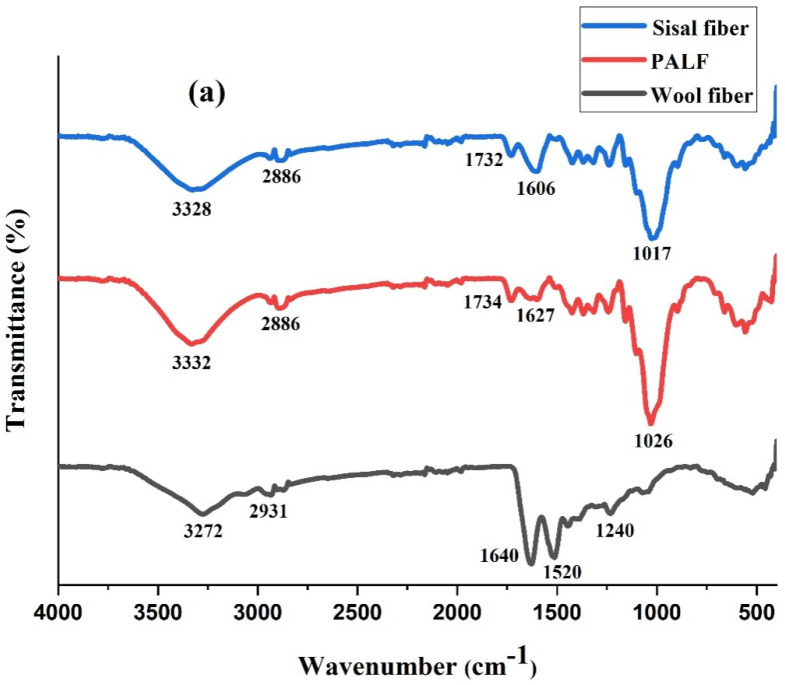
FTIR spectrum of (**a**) sisal fibre, PALF and wool fibre (**b**) vulcanized rubber, PWHC and SWHC.

**Figure 4 polymers-14-04882-f004:**
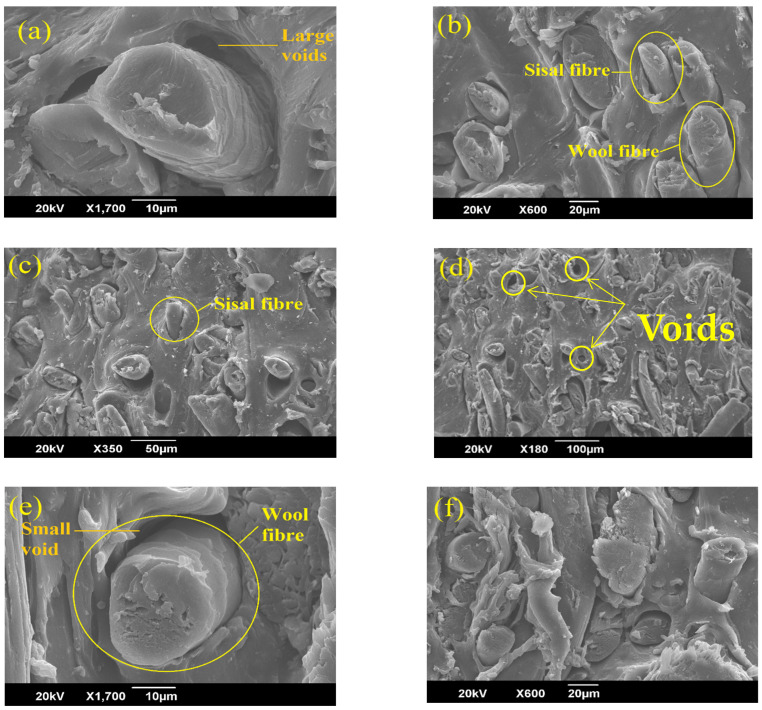
SEM images of the SWHC (**a**–**d**) and the PWHC (**e**–**h**) at various magnifications.

**Figure 5 polymers-14-04882-f005:**
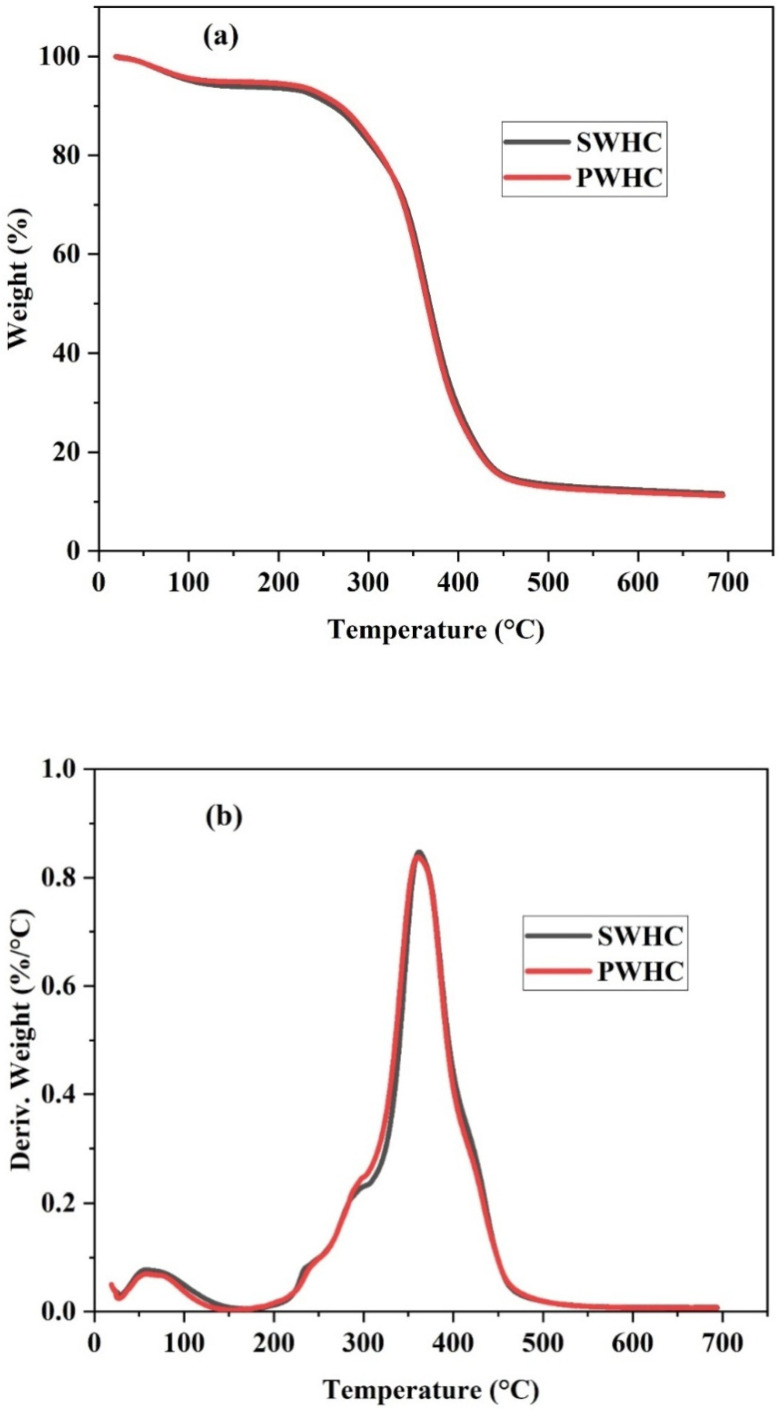
(**a**) TGA and (**b**) DTG curve of the SWHC and PWHC.

**Figure 6 polymers-14-04882-f006:**
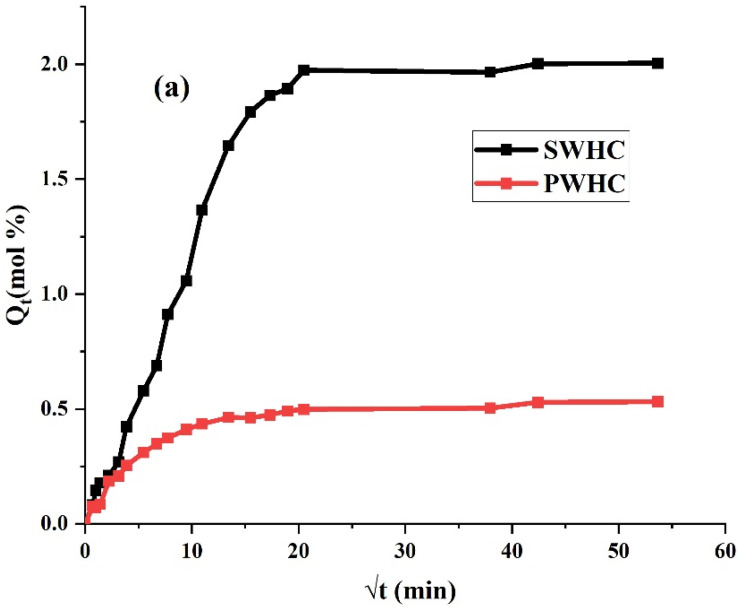
Diffusion curve of the SWHC and PWHC in (**a**) toluene (**b**) water.

**Figure 7 polymers-14-04882-f007:**
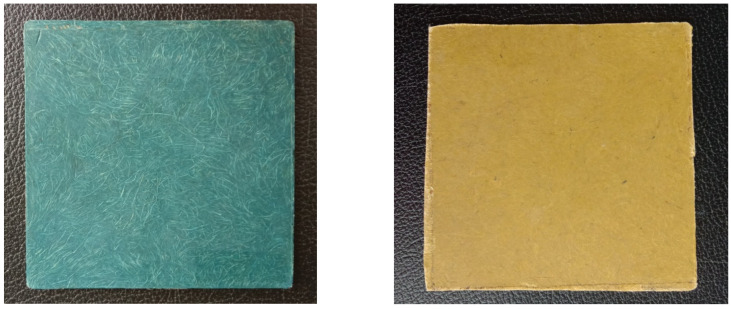
Images of the SWHC (**green**) and the PWHC (**yellow**). Note: Pigments were added during the composite preparation for identification.

**Figure 8 polymers-14-04882-f008:**
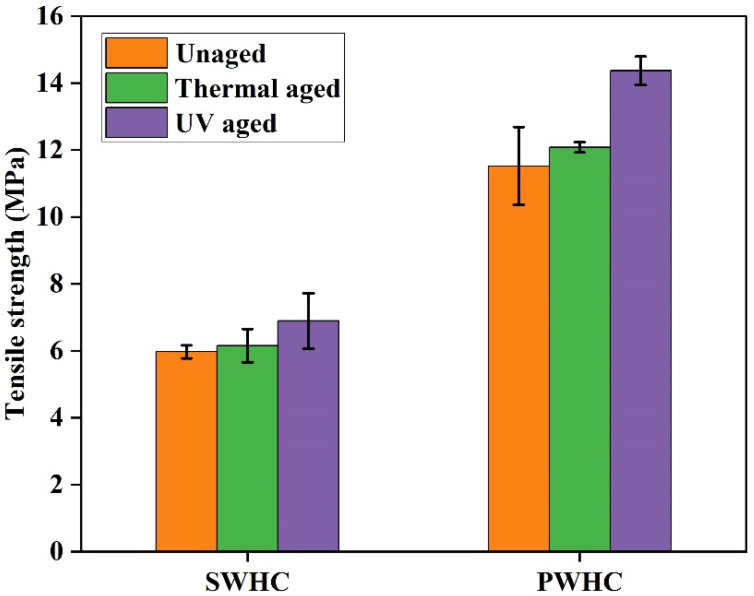
Mechanical properties of the SWHC and PWHC after the thermal and UV aging.

**Table 1 polymers-14-04882-t001:** List of rubber compounding ingredients [19,20].

Ingredients	SWHC (phr)	PWHC (phr)
Natural Rubber	100	100
Zinc Oxide	5.0	5.0
Stearic Acid	2.5	2.5
Wingstay L	1.0	1.0
CBS	1.5	1.5
Wool Fibre	50	50
Sisal Fibre	50	-
PALF	-	50
Sulphur	2.5	2.5

phr indicates parts per hundred of rubber.

**Table 2 polymers-14-04882-t002:** Torque and cure time values of the SWHC and PWHC samples.

Sample Name	Maximum Torque (dNm)	Minimum Torque (dNm)	t_90_ (min)	ts_2_ (min)	Cure Rate Index (min^−1^)
SWHC	38.36	0.76	5.12	2.29	35.34
PWHC	48.68	1.72	5.14	2.29	35.09

**Table 3 polymers-14-04882-t003:** Tensile and tear properties of the SWHC and PWHC.

Sample	Tensile Strength at Break (MPa)	Elongation at Break (%)	Young’s Modulus (MPa)	Tear Strength (N/mm)
SWHC	6.09 (3.30)	5.42 (8.85)	169.3 (1.36)	45.9 (6.48)
PWHC	11.14 (7.08)	4.49 (3.1)	334 (3.00)	85.0 (6.18)

**Table 4 polymers-14-04882-t004:** Moisture content and hardness of the PWHC and SWHC.

Sample	Moisture Content (%)	Hardness (Shore A)	Areal Density (g/m²)	Relative Density (g/cm^3^)
PWHC	5.90	91.56	2719.36	1.11
SWHC	6.57	91.30	2707.84	1.09

**Table 5 polymers-14-04882-t005:** Crosslink density and Mc of the SWHC and PWHC.

Composite	Mc (g/mol)	Crosslink Density (mol/g)
SWHC	62,840.43	7.96 × 10^−6^
PWHC	65,247.92	7.66 ×10^−6^

**Table 6 polymers-14-04882-t006:** Weight reduction of the SWHC and PWHC after 60 days of soil burial test.

Sample	Weight Reduction (%)
SWHC	2.43 (0.38)
PWHC	2.34 (0.49)

Note: The values in the parenthesis indicate the standard deviation.

## Data Availability

Not applicable.

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
