# Peer review of "Physico-Mechanical, Thermal, Morphological, and Aging Characteristics of Green Hybrid Composites Prepared from Wool-Sisal and Wool-Palf with Natural Rubber"

_polymers, 2022, doi:10.3390/polym14224882_

Round 1
Reviewer 1 Report
This manuscript reports Sisal-Wool hybrid composite and Pineapple Leaf fiber-wool hybrid composite. Various properties of SWHC and PWHC were compared. OWHC showed higher physical mechanical properties SWHC. The manuscripts provide many useful data, well organized, and easy to read. The manuscripts will be improved if following points are addressed.
1. Line 27 and Line 31, You are using different referencing style.
2. In Figure 1, briefly state why the torque decrease after the maximum value.
3. Please give information about PALF and sisal fiber such as fiber diameter, density, etc. Also please give chemical structure of PALF and sisal as a reference for FT-IR study in Figure XX. It’s better to provide SEM images of wool, sisal, PALF fibers.
4. The authors mentioned that the objective of this work is to give a value addition to highly coarse wool. I think it is better to compare data of SWHC and PWHC with those of Wool only-rubber composite.
5. Line 247, I cannot see Figure 3(b)
6. About Figure 4, it is said that PWHC shows smaller voids and higher packing density than SWHC. But, I cannot see these observations in the SEM images. Please exactly indicate void and packing in the figures and compare PWHC and SWHC.
7. In Figure 4, how do you know which one is PALF, sisal fiber, or wool fiber? Is it based on the fiber diameter? Diameters look very similar.
8. In Figure 7, why do you think that elongation of UV aged SWHC increased?
Author Response
- Line 27 and Line 31, You are using different referencing style Reply - The consistency of referencing style has been maintained in the revision
- In Figure 1, briefly state why the torque decrease after the maximum value. Reply - The torque is getting decreased due to the over curing of rubber. This has been explained with citation.
- Please give information about PALF and sisal fiber such as fiber diameter, density, etc. Also please give chemical structure of PALF and sisal as a reference for FT-IR study in Figure XX. It’s better to provide SEM images of wool, sisal, PALF fibers. Reply - The fibre diameter and density are incorporated in the revision.
Sisal and PALF are ligno cellulosic fibre. Chemical structure is same, and they chemically differs with the % of cellulose, hemi cellulose and lignin.
Due to the long queue for getting the slot for SEM images of the fibres, we couldn’t add the SEM images. The SEM images of the fibres are already published in the below articles. Out of which two articles are our own research work.
10.1016/j.indcrop.2021.114489
10.1016/j.jclepro.2016.09.092
10.1080/15440478.2011.551002
- The authors mentioned that the objective of this work is to give a value addition to highly coarse wool. I think it is better to compare data of SWHC and PWHC with those of Wool only-rubber composite. Reply - Totally agree with reviewers comment. The said experiment has been done and this manuscript has been communicated for publication. The results are as follows. Tensile strength – 10.6 MPa, Elongation – 6.9 %, Modulus – 193 MPa
- Line 247, I cannot see Figure 3(b) Reply - We really wonder how it happened. The Figure 3 b was there while uploading the manuscript, but not visible in the manuscript, which is given for revision. We kept Fig3.b in the revision as such.
Figure 3 has been split into (a) and (b) part where (a) contains FTIR spectrum of fibers while (b) contains that of vulcanized rubber and composites.
-
About Figure 4, it is said that PWHC shows smaller voids and higher packing density than SWHC. But, I cannot see these observations in the SEM images. Please exactly indicate void and packing in the figures and compare PWHC and SWHC. Reply - The void around the protruding fibre is less in PWHC more in SWHC. It is marked in SEM images in the revision.
Enlarged SEM images of PWHC and SWHC are attached as supplementary file. The voids are apparent in the images.
- In Figure 4, how do you know which one is PALF, sisal fiber, or wool fiber? Is it based on the fiber diameter? Diameters look very similar. Reply - Wool fibre is having typical scale structure on its surface which is marked in the SEM images. Thus, it is easy to identify the wool fibre in the composite. However, it is very difficult to distinguish sisal and PALF since both of them are having almost same surface characteristics. But, sisal possesses higher fibre diameter than PALF. The fineness of both fibres are included in the revision.
-
In Figure 7, why do you think that elongation of UV aged SWHC increased? Reply -
We also noticed the same. Generally the after ageing, due to the increase in the cross linking, the elongation properties used to decrease. Contradictorily, in the case of SWHC, the elongation properties were increased. The below research article says that aging causes a breakdown of some bonds between polymer chains allowing slip movement between molecules. The ease of this movement can be seen from the increased vulcanized rubber elasticity. The following articles reported the same.
Doi: 10.31940/logic.v19i3.1469

Reviewer 2 Report
The subject of this manuscript is important to readers thus I recommend accepting it after the below comments:
1- Please move all findings from the abstract to the conclusion. 2- Modify the introduction, focus more on your area, and add the latest studies. 3- Improve the result and discussion.Author Response
Please move all findings from the abstract to the conclusion.
Modify the introduction, focus more on your area, and add the latest studies.
Improve the result and discussion.
Reply - Many thanks for the comments. We modified the manuscript accordingly
Reviewer 3 Report
polymers-2011928
Physico-mechanical, thermal, morphological, and aging characteristics of green hybrid composites prepared from wool – sisal 3 and wool - PALF with natural rubber
Good meticulous work for present day research; especially for packing and household applications. The authors have done a worthful study in comparing the curing characteristics, physico-mechanical properties, solvent diffusion properties and accelerated aging of plant leaf fibre - animal fibre hybrid composites.
For a curiosity, the author should answer for the below questions:
Ø Will there be any impact on chemical structure of NR? If yes, explain with validation.
Ø What will be the effect, if the fibers are pretreated?
Ø What is the purpose of adding coarse wool fiber in both the hybrid composites? Will they improve any compatibility?
Ø What made the author to compare plant and animal-based fiber in this study? Is there any logic behind this selection?
The results and discussions are acceptable and well-presented. Include the below references to add more value for the publication; https://doi.org/10.1533/9780857096913.2.249, https://doi.org/10.24425/amm.2021.136395, https://doi.org/10.3390/fib9020011. The technical depth is very much appropriate for the general knowledgeable individuals working in the field of natural fiber composites. I, as a reviewer of this manuscript, will accept this quality manuscript for publication in “Polymers”.
Author Response
Will there be any impact on chemical structure of NR? If yes, explain with validation.
Reply - From the FTIR analysis, we found no chemical interaction with NR with either of the fibres. NR is non polar and the fibres are polar. It is just acting as a matrix for the fibres. Thus, we assume, there is little impact on chemical structure of NR in the developed hybrid composites.
What will be the effect, if the fibers are pretreated?
Reply - There is a possibility of better interaction with NR with cellulosic natural fibres if pretreated with NaOH or enzymes. Few studies are already reported.
https://doi.org/10.1016/j.compositesa.2005.03.004
https://doi.org/10.1016/j.egypro.2014.07.203
In case of wool, since it is sensitive to alkali, instead of alkali pre-treatment, enzyme treatment would be favoured.
In our previous study, we found that the treatment of wool with Sodium ligno sulphonate increases the compatibility with natural rubber
https://doi.org/10.1016/j.indcrop.2021.114489
What is the purpose of adding coarse wool fiber in both the hybrid composites? Will they improve any compatibility?
Reply - The reported work is a part of PhD dissertation based on coarse wool- Polymer composites. This work is reported for the first time. Initially, we developed coarse wool- NR composites. (This manuscript has been communicated for publication). However, we realized the need of improving the properties of the wool- NR composite. For this purpose, hybrid composites were developed.
What made the author to compare plant and animal-based fiber in this study? Is there any logic behind this selection?
Reply - Kindly refer the answer for the previous query
The results and discussions are acceptable and well-presented. Include the below references to add more value for the publication; https://doi.org/10.1533/9780857096913.2.249, https://doi.org/10.24425/amm.2021.136395, https://doi.org/10.3390/fib9020011. The technical depth is very much appropriate for the general knowledgeable individuals working in the field of natural fiber composites. I, as a reviewer of this manuscript, will accept this quality manuscript for publication in “Polymers”.
Reply - Thank you very much for your valuable suggestions and encouraging comments. We have included the relevant referred citations.